# Single-Particle Tracking of *Thermomyces lanuginosus* Lipase Reveals How Mutations in the Lid Region Remodel Its Diffusion

**DOI:** 10.3390/biom13040631

**Published:** 2023-03-31

**Authors:** Josephine F. Iversen, Søren S.-R. Bohr, Henrik D. Pinholt, Matias E. Moses, Lars Iversen, Sune M. Christensen, Nikos S. Hatzakis, Min Zhang

**Affiliations:** 1Department of Chemistry & Nanoscience Center, University of Copenhagen, Thorvaldsensvej 40, 1871 Frederiksberg, Denmark; 2Novo Nordisk Foundation Centre for Protein Research, Faculty of Health and Medical Sciences, University of Copenhagen, Blegdamsvej 3B, 2200 Copenhagen, Denmark; 3Department of Physics, Massachusetts Institute of Technology, Cambridge, MA 02139, USA; 4Novozymes A/S, 2800 Kgs. Lyngby, Denmark

**Keywords:** *Thermomyces lanuginosus* lipase, lid mutations, application condition, single-particle tracking

## Abstract

The function of most lipases is controlled by the lid, which undergoes conformational changes at a water–lipid interface to expose the active site, thus activating catalysis. Understanding how lid mutations affect lipases’ function is important for designing improved variants. Lipases’ function has been found to correlate with their diffusion on the substrate surface. Here, we used single-particle tracking (SPT), a powerful tool for deciphering enzymes’ diffusional behavior, to study *Thermomyces lanuginosus* lipase (TLL) variants with different lid structures in a laundry-like application condition. Thousands of parallelized recorded trajectories and hidden Markov modeling (HMM) analysis allowed us to extract three interconverting diffusional states and quantify their abundance, microscopic transition rates, and the energy barriers for sampling them. Combining those findings with ensemble measurements, we determined that the overall activity variation in the application condition is dependent on surface binding and lipase mobility when bound. Specifically, the L4 variant with a TLL-like lid and wild-type (WT) TLL displayed similar ensemble activity, but WT bound stronger to the surface than L4, while L4 had a higher diffusion coefficient and thus activity when bound to the surface. These mechanistic elements can only be de-convoluted by our combined assays. Our findings offer fresh perspectives on the development of the next iteration of enzyme-based detergent.

## 1. Introduction

Lipases (triacyl glycerol ester hydrolase, E.C. 3.1.1.3) catalyze the hydrolysis of triglycerides to glycerol and free fatty acids, and are one of the most promising kinds of enzymes used in biocatalysis. Most lipases become fully activated upon binding to a water–lipid interface, a mechanism called interfacial activation [1,2]. This is manifested by the “lid” region that covers the active site; upon interaction with the substrate interface, it rotates around its hinge and exposes the catalytic site [3,4]. In the absence of a substrate interface, lipases display very low activity for water-soluble substrates. One prominent example is *Thermomyces lanuginosus* lipase (TLL). Studies indicated the lid in a closed or open state is crucial for TLL’s activation [4]. Alteration of the lid residue of TLL has been shown to have a significant influence on interfacial activation and enzymatic activity [5,6].

One important application of TLL and related lipases is the enzyme-based detergent industry. Enzyme activity depends on the chemical environment in detergents, including solvent polarity, pH, ion effects, and surfactants. To improve the efficiency of TLL in the detergent industry, it is necessary to first understand the functions of TLL lid mutations in application conditions, and implement this to rationally design new improved variants.

Current understanding of lipase performance in application conditions mainly relies on spectra evidence reporting average kinetics of large ensembles of enzymes in solution, masking the heterogeneousness of protein activities which are crucial for the underlying regulation of biomolecular surface recognition and function of the protein [7,8]. Through the ability to directly observe enzyme conformational sampling and function dynamics, single-molecule techniques have become a powerful tool for the study of the heterogeneity of enzyme conformations and their association with various protein functional states [9,10,11,12,13]. Single-particle tracking (SPT) assay, as an important component of single-molecule techniques, allows the analysis of diffusional properties of individual molecules, providing insights into protein functions [14,15,16,17,18]. In our previous work, we successfully established an SPT assay platform to directly observe diffusion behaviors of individual TLL molecules and investigated how variants affect this on a clean biophysical system with trimyristin as the clean substrate [14]. Lipase variant 1 (L1) contains the entire lid of a ferulic acid esterase from *Aspergillus niger* (FAEA), which is a lipase structurally similar to TLL, while attaining an open lid conformation in the solution, and therefore, is active on water-soluble substrates [19,20]. The L2 variant contains an FAEA-like lid with the N-glycosylation site removed. Thus, L1 and L2 are hypothesized to have more open conformation and have shown much lower activities than wild-type (WT) TLL [6,21]. The L4 variant contains the entire lid domain from *Aspergillus terreus* lipase (ATL), with similar hydrophobic residues to those of WT TLL [6]. As a result, L4 and WT are proposed to have TLL-like characteristics. In this work, we employed the SPT assay platform to directly observe for the first time the temporal trajectories of WT TLL and the three TLL variants (L1, L2, and L4) [6] on a natural substrate surface and attained a quantitative understanding of how lid mutations affect behavior in application conditions. 

To evaluate lipase behavior in application conditions, we used lard, the animal fat derived from pigs that has been widely used to test lipase performance in enzyme and detergent industries [22,23,24], as a natural-like substrate. The application condition we used, a mixture of anionic linear alkylbenzene sulfonate (LAS) and nonionic hexamethylene glycol monododecyl ether (HE) surfactants commonly found in household detergents, contained calcium ions and surfactants. Thousands of individual enzyme behaviors for each type of TLL variant were observed using TIRF microscopy. Quantitative analysis of the kinetics using hidden Markov modeling (HMM) revealed three diffusional states and the corresponding lifetime and rates for each pair of transitions. As a result, it allowed us to extract the energy barriers that describe the diffusion behaviors of WT and TLL variants in the application condition. Our results revealed the effect of the lid mutations of TLL in the application condition. Although they had the same number of diffusional states, the four types of TLLs displayed significantly different transition pathways. The findings provide new insights into the design of the next generation of enzyme-based detergent.

## 2. Materials and Methods

**Materials.** Chemicals such as 1,2-Di-O-lauryl-rac-glycero-3-(glutaric acid 6-methyl- resorufin ester) (lipidated resorufin) (CAS: 195833-46-6), Trizma base (TRIS) (CAS: 77-86-1), HEPES ≥ 99% (titration) (CAS: 7365-45-9), n-Hexane (CAS: 110-54-3), and calcium chloride (CAS: 10043-52-4) were of analytical grade and purchased from Merck (Søborg, Denmark) unless otherwise stated. TLL variants L1, L2, L4, and WT were provided by Novozymes A/S, Denmark. SeTau-647-Maleimide was purchased from SETA BioMedicals. Labeled phospholipid 1,2-Dioleoyl-sn-glycero-3-phosphoethanolamine-Atto488 (DOPE-ATTO488) was purchased from ATTO-TEC GmbH, Siegen, Germany. Surfactant 52044- hexaethylene glycol monododecyl ether (HE, CAS: 3055-96-7) was purchased from Nikko Chemicals Co., Ltd., Tokyo, Japan, and linear alkylbenzene sulfonate (LAS) was provided by Novozymes A/S, Denmark. Lard was purchased from Dragbæk A/S, Denmark.

**Protein engineering and purification.** Protein engineering and purification were performed as previously described [6]. Specifically, FAEA has similar tertiary structures to TLL but different activation functionalities. Given the distribution of polar residues, including an N-glycosylation site in its lid, the activity of FAEA is independent of an interface. To change the activation mechanism in TLL, a rational design approach was applied on the basis of the lid residue composition in FAEA. Variants L1 and L2 contain FAEA-like lids. L4, a TLL-like variant that contains the entire lid domain from ATL, was included as a proof of concept demonstrating the impact of lid residue composition on activation.

The protein purifications were carried out using ÄKTA Prime equipment (Amersham Biosciences). The high purity of variants was confirmed by SDS-PAGE, with no contamination observed.

**Lipase labeling.** WT and the three TLL variants were labeled with SeTau 647 maleimide (Thermo Fisher Scientific) on the free cysteine C137 site following the manufacture’s protocol [25]. Free dyes were removed by Amicon filters. Labeling efficiencies were characterized by Nanodrop. The labeled enzymes were stored in 1.2 μM aliquots of 20 μL at −80 °C. Each aliquot was only used once.

**The application system.** To simulate the common washing conditions, we added CaCl_2_ and a surfactant mixture containing a 1:2 mol% ratio of HE and LAS to 50 mM TRIS buffer at pH 7.8, and the final concentrations of calcium and surfactants were 3 mM and 500 μM, respectively. Lard (pig fat) was used as the substrate. 

**Ensemble activity assay.** The lipase activity assay was performed on an Infinite® M1000 PRO plate reader (Tecan, Männedorf, Switzerland). A mixture of 10 μM Resorufin Lipase substrate and 2 mM LARD in hexane was dispensed carefully in black 96-well Microtiter plates (CAT: 237107, Thermo Fisher Scientific, Roskilde, Denmark) with 200 μL in each well. The hexane was evaporated in a fume hood for at least 4 h in the dark, minimizing the bleaching of lipidated resorufin. Lipase was dissolved in 50 mM TIRS buffer, containing 3 mM CaCl_2_ and 500 μM surfactants (HE:LAS = 1:2 mol%) at pH 7.8. Then, 200 μL lipase solution was added to each well at a concentration of 1.25 ppm. Lipases catalyzed hydrolysis of substrates, resulting in resorufin cleavage from the substrate surface, which was made fluorescent by excitation with a 530 nm laser. The increase in fluorescence intensity due to the release of resorufin analog was measured every minute for one hour with excitation at 530/10 nm, emission at 590/10 nm, and gain at 1200. 

**Lipase binding assay.** The binding of enzymes to the substrate surface was measured on a confocal microscope (IX83, Olympus, Tokyo, Japan) with a 640 nm OBIS laser line (COHERENT, Dieburg, Germany). The substrate surfaces were prepared the same way as the ensemble activity measurements. An image series of SeTau 647-labeled WT and TLL variants binding to the substrate surface in the application condition was acquired with a CMOS camera (Olympus DP74, Tokyo, Japan) using a 10× objective (Olympus UPLSAPO 10X2, Tokyo, Japan). The mean intensity of each frame was plotted versus time t (30 min in total) to extract the amount of TLL docked on the surface.

**Substrate surface preparation for SPT.** Lard substrate surfaces were prepared using an in-house-developed method [14], where14 μL toluene solution containing 24.5 g/L LARD with 0.05 ppm DOPE-ATTO488 dissolved in toluene was spin-coated on Ø25 mm round microscope glass slides at 5000 rpm for 2 × 60 s with a 10 s pause in-between. The spin-coated glass slides were assembled in custom Teflon chambers and subsequently kept in a high vacuum for at least 2 h before immediate use.

**Total internal reflection fluorescence (TIRF) microscopy.** A TIRF microscope (IX83, Olympus) was used for the single-particle tracking (SPT) experiments. Oil immersion objective (UAPON 100XOTIRF, NA 1.49, Olympus) and an EMCCD camera (ImagEM X2, Hamamatsu, Shizuoka, Japan) were used to record images and videos with a pixel width of 160 nm and field of view 81.92 μm × 81.92 μm. Laser lines of 488 nm and 640 nm were used to excite the fluorophores ATTO-488 (substrate surface) and SeTau 647 (enzymes), respectively. Imaging was performed with an exposure time of 80 ms, 100 nm penetration depth, and 300 EM gain. Each image series contained 50 frames of the 488 nm channel (15% laser power) and then 2000 frames of the red channel (10% laser power). Prior to imaging, 53 μL 50 mM TRIS buffer containing 3 mM CaCl_2_, pH 7.8 was added to the chamber, followed by a background image series, which allowed for individual surface background correction. Then, 2 μL labeled lipases were added to the chamber, resulting in a final concentration of ~0.5 nM. Finally, 5 μL the surfactant mixture was added to reach the concentration of 500 μM.

**Image analysis and diffusion coefficient extraction.** Quantitative image analysis was performed using TrackPy [26] together with an in-house-developed Python script [14,27]. 

To determine the localization error of each type of TLL, we first calibrated EMCCD from background measurement to convert image pixel values to photon counts, then identified fluorescent spots in the movie and cropped boxes around point spread functions (PSFs). Gaussian fitting the PSFs allowed us to extract the photon count per PSF, width, and background photon count. We generated PSFs with the fitted parameters using the fitted EMCCD model and used TrackPy to estimate the center and calculate the error from the true value. We determined the localization error for each measured sample by plotting the distribution of errors and calculating the average (Appendix A). 

To extract the diffusion coefficient, we assumed the particles have Brownian diffusion and applied a method to describe the data [14,18], where the probability at a given step length *r* is given by
$$p\left(r,t,D\right)=\frac{r}{2Dt}*exp(-\frac{r^{2}}{4Dt})$$
where *r* is the observed step length, *t* is the time between consecutive steps, and *D* is the diffusion coefficient determined by the maximum likelihood approach, which has no need for binning the data and thus avoids any binning bias.

**Resolving diffusional states**. TLL diffusional states were resolved by a hidden Markov model (HMM) approach, similar to previously published methodologies [14,27,28]. HMM analysis was performed on the distribution of observed step lengths for each trace followed by a gamma distribution. Analysis of traces across WT and TLL variants via Bayesian information criterion (BIC) score [29] revealed that a 3-state model best described TLL mobility under the application conditions. The total distribution of observed step lengths for WT and each TLL variant was fitted by 1-4 gamma distribution(s). All found states, for a given lipase type, were plotted together in a histogram and fitted with a mixture of three Gaussians.

**Transition rates and energy determination**. Each pair of mobility transitions (“state before” and “state after”) was plotted in transition density plots (TDPs) and subsequently separated into clusters by combining k-means clustering and a two-dimensional Gaussian mixture model. Transition rates were determined by fitting a single exponential decay to the lifetime of each cluster. The energy barrier *E_a_* for each transition and the relative free energy difference between adjacent states were determined by transition state theory:
$$E_{a}=-RT\;ln(\frac{hk_{ij}}{k_{B}T})$$
$$\Delta G=-RT\;ln(\frac{k_{ij}}{k_{ji}})$$
where *R* is the gas constant, *T* is the temperature in Kelvin (298 K), *h* is Planck’s constant, *k_ij_* is the transition rate from *i* to *j*, and *k_B_* is the Boltzmann constant.

## 3. Results and Discussion

### 3.1. Single-Particle Tracking of WT and TLL Variants

We used an SPT assay to directly observe the binding and diffusional behaviors of individual WT and TLL variants on a lard substrate surface in the application condition (Figure 1a). TLL enzymes were specifically labeled by SeTau647 on a single cysteine site (D137C). The high quantum yield, high photostability, and long lifetime of SeTau647 [30,31] allowed us to record the diffusional behavior of thousands of individual TLL enzymes in parallel for a longer time compared to the SPT performed in our previous work [14]. The effect of labeling on lipase activity was tested by an ensemble activity assay. Similar activity was seen between WT-non labeled and WT-labeled with SeTau647 (Figure 2a), demonstrating that SeTau 647 labeling does not affect the function of TLL enzymes. 

Thousands of TLL trajectories on a lard surface were recorded in parallel by a TIRF microscope with 81 ms temporal resolution. We first precisely localized particle positions via two-dimensional Gaussian fitting from time-series images as we routinely do [14,18,27,32,33,34,35,36], and consequently, spatiotemporal trajectories by connecting particle coordinates between frames. We then extracted the distance that an enzyme traveled throughout the observation time (step length). The recorded tracks (see the zoom in Figure 1b for an example), as well as the step length traces of TLL enzymes (Figure 1c and Appendix A), revealed heterogeneous mobility behaviors. 

The mobility of enzymes is suggested to correlate with activity, and we recently showed evidence supporting this for lipase [14,18]. Generally, lipases with high ensemble activities display a high average diffusion coefficient on the substrate surface. Here, lipase molecules’ diffusion behaviors were described by Brownian motion. We utilized maximum likelihood estimation (MLE) to fit the step length distribution per trajectory and consequently determined the average diffusion coefficient of an enzyme (Section 2) [14,18,27]. The diffusion coefficient allowed us to evaluate the catalytic activity of individual enzymes on the substrate surface.

**Figure 1 biomolecules-13-00631-f001:**
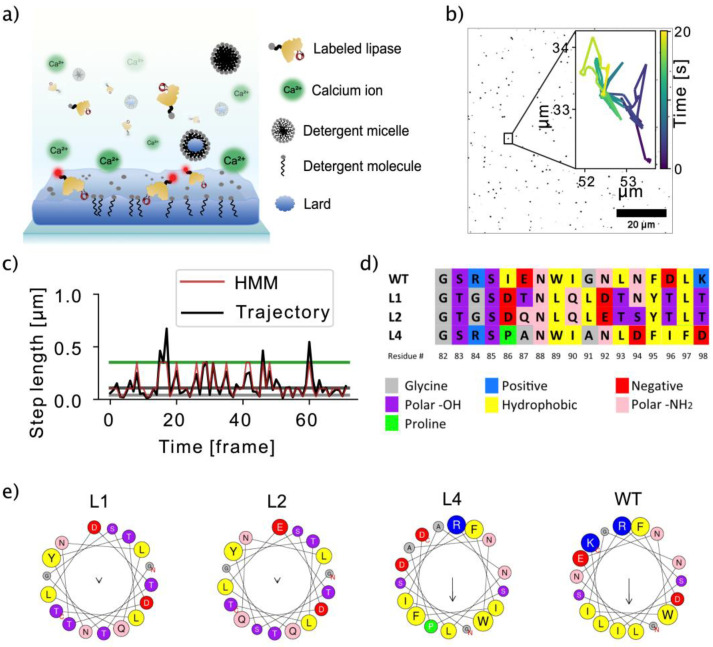
SPT assay of lipase activities in application conditions. (**a**) Schematic illustration of the experimental setup. Diffusion of SeTau 647-labeled TLL enzymes on a lard surface was investigated by TIRF microscopy in the presence of calcium ions and surfactants. (**b**) A representative micrograph displaying individual TLL molecules (black dots) bound to the lard surface. The zoom is a temporal trajectory of a single TLL variant’s diffusion, which is color-coded according to observation time. (**c**) Typical step length traces showing the reversible transition between different states, along with the corresponding idealistic traces discovered via HMM analysis. (**d**) Amino acid sequence alignment of lid region for WT and the three variants used in this work. The charge, polarity, and type of amino acid are indicated by color coding. (**e**) The lid structures are displayed as a helical wheel generated by HELIQUEST [37].

**Figure 2 biomolecules-13-00631-f002:**
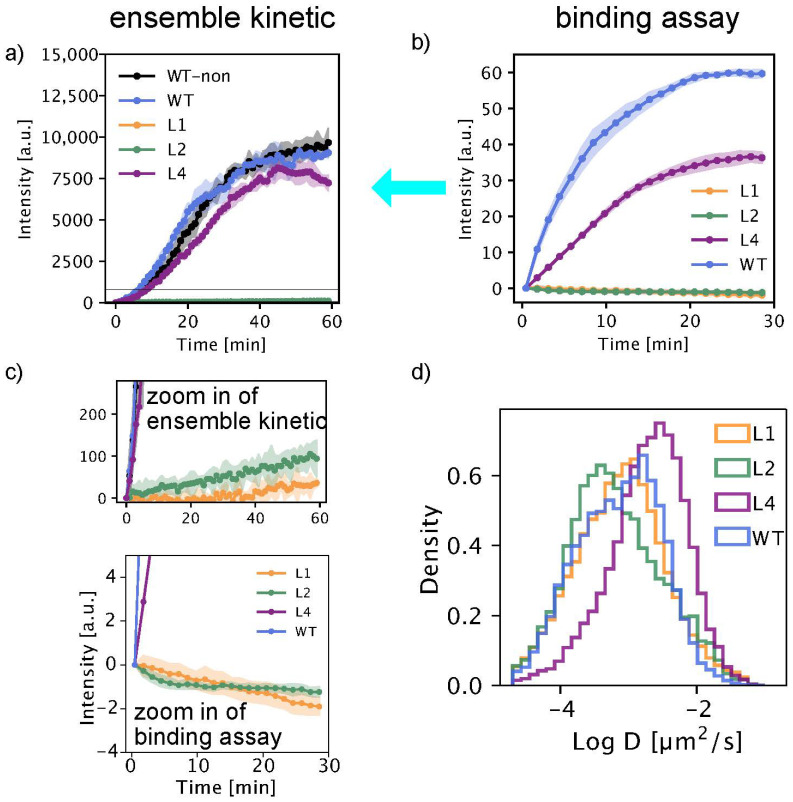
(**a**) Ensemble activity of WT and TLL variants in the detergency application system. (**b**) Binding assay of WT and TLL variants in the detergency application system. (**c**) Top: zoom in on the ensemble activity data in (**a**). Bottom: zoom in on the binding data in (**b**). The negative binding values of L1 and L2 are due to the first surface binding intensity being set to zero. (**d**) Quantification of the mobility of individual lipase types on lard surfaces in the application system. Histogram plots of the distribution of lipase as the logarithm of the diffusion coefficients (log D) across the four types of TLLs.

### 3.2. WT and TLL Variants in the Detergency Application System

The lid structures of WT and the three TLL variants used in this work are shown in Figure 1d,e [6,21]. The ensemble activities of TLL-like variant L4 and WT were both significantly increased in the application condition (Figure 2a,c). In addition, the binding efficiencies revealed that both WT and L4 had significantly higher binding to the surface than the rest of the variants (Figure 2b,c). This suggests that, at least in this case, the enhanced activity of WT and L4 can be attributed to their high binding efficiencies in the application condition. Calcium ions in the solvent can lower the dielectric constant at the water–lipid interface [38]. The lid region of TLL-like lipase displays high calcium dependency [6,38], thus resulting in a markedly enhanced binding and activity of WT and L4. It is interesting to see that although the binding efficiency of WT was obviously higher than L4, their ensemble activities were quite similar. After careful inspection of the diffusion coefficients of WT and all TLL variants, we found that the diffusion coefficient of WT on lard surfaces was surprisingly lower than L4, with averages of 0.0518 and 0.0764 μm^2^/s, respectively (Figure 2d and Appendix A). This indicates calcium could not affect lipase mobility after they bound to the substrate surface [18], which is consistent with earlier studies showing that the catalytic activities of TLL were inhibited by surfactant below its critical micelle concentration [39]. The methodology presented here allowed us to deconvolute the effect of binding to the surface and the activity when bound on the surface that is averaged out in conventional assays. This highlights the power of our SPT assay in deciphering the interface differences of TLL variants, which is crucial for the design of new TLL variants.

Unlike the effect on TLL-like variants L4 and WT, calcium ions have almost no effect on the FAEA-like variants [6], resulting in L1 and L2 displaying low activities and bindings, just like in the buffer conditions. This is expected with their FAEA-like lid properties, which are less hydrophobic, leading to the incorrect orientation of the enzyme on the lipid interface or imperfect active site organization, as previously suggested [14]. The relatively lower diffusion coefficients of L1 and L2 on the substrate surface (Figure 2d and Appendix A) further confirmed the fact that lipases’ mobility correlates with their activity [14,18].

### 3.3. Three Diffusion States of WT and TLL Variants in the Application System

Careful inspection of the step length traces in Figure 1c and Appendix A indicated that TLL molecules sample multiple mobility behaviors. Here, we utilized HMM based on our recently published method [14,18,27] to further explore the mobility differences among the four types of TLLs. As shown in Figure 3a and Appendix A, WT and all TLL variants showed good agreement with a three-state model, where three clear distributions in idealized step length were displayed for all after HMM segmentation, corresponding to an arrested, practically immobile state, a slow state with diffusion slightly faster than the immobile one, and a fast diffusional state which is significantly faster than the other two states. All four types of TLLs showed basically the same step lengths in the immobile state, with a mean value of around 0.03 μm, which we showed to be close to the localization error [14,18]. The mean step size of L4 in the slow state was around 0.1 μm, which appeared larger than the ~0.09 μm of the other three TLLs. Meanwhile, in the fast state, which contributed the most to activity, the four types of TLLs exhibited considerably different step-length distributions. As the FAEA-like lids of L1 and L2 are less flexible than the TLL-like lid, the step sizes of L1 and L2 in the fast state were smaller than WT and L4. Combining their occupancy in the three states (Figure 3b), we found L1 and L2 to be the lower activity-variants, which spent 7% of their time in the fast state and ~30% in the immobile state. Meanwhile, the higher activity variant L4 spent a relatively long time in the fast state (8%) and ~20% in the immobile state. Although WT displayed a longer step length in the fast state, it had the lowest occupancy in the fast state (6%) as well as the highest occupancy in the immobile state (38%). Earlier studies suggested small molecules of surfactants fit into the active site cleft and inhibit the catalytic activity of WT by sterically blocking the substrate [39], thus resulting in the relatively low activity of WT. Deciphering this extends beyond the scope of this work but clearly deserves further investigation in a future publication.

To further investigate the pathways connecting TLL states and the relative frequencies of state transitions between the three states in different types of TLLs, we used transition density plots (TDP) of traces segmented by HMM (Appendix A). TDP allows us to extract the transition lifetime and the corresponding rates for all pairs of transitions (immobile ⇋ slow ⇋ fast ⇋ immobile) (see Appendix A for the lifetime, Figure 3b and Appendix A for the rates). Subsequently, the energy barriers for all transitions can be calculated via a combination of k-means clustering and a two-dimensional Gaussian mixture model followed by a fit of the decaying lifetime of transitions (see Section 2 and Appendix A). As expected, the transition rates and energy barriers vary with TLL types, ranging from 0.27 to 2.48 s^−1^ and 70.7 to 76.2 kJ/mol, respectively, consistent with our previous studies [14,18]. For WT and all TLL variants, the transition energies to the slow state are lower than those from the slow state, suggesting the slow state has the lowest free energy and is the most thermodynamic stable state in the application condition. This explains the high occupancy of the slow state (Figure 3b). The higher number of recorded events (16,972 traces here for the WT vs. 4077 traces in our previous work [18]) allowed us to extract the transition between the immobile and fast states with statistical significance. It can be seen that all types of TLLs show a low transition frequency between the immobile and fast states in the application condition. The transition rate of WT lipase from fast to slow state is much faster than the three variants, and the WT has the lowest transition frequency to the fast state (Appendix A), resulting in its low occupancy of the fast state as well as the low average diffusion coefficient. The highly active variant L4 has higher transition frequencies between the slow and fast states than the other TLLs. The transition behaviors of L1 and L2 are quite similar (except L1 displays a higher transition rate from the fast to immobile state) and are in agreement with their performance under buffer conditions [6,14], further indicating their activities are mainly controlled by the lid mutations.

## 4. Conclusions

The key role of the lid in regulating substrate specificity and the catalytic mechanism of lipases has been emphasized in a number of studies [6,40,41]. Due to the heterogeneous mobility behaviors of lipases at the substrate interface, traditional characterization methods such as spectroscopy that provide only the average results are not sufficient for lipase study. Therefore single-molecule techniques are highly desired. In this work, we used SPT to directly observe the individual behaviors of WT and TLL variants in an application system. HMM analysis was utilized to identify the three diffusional states and their corresponding abundance associated with lipase function. The parallelized nature and high temporal resolution (81 ms) of SPT combined with ensemble activity assay and surface binding assay allowed us to decipher the effect of lid mutations on lipase activity in the application conditions. 

The ensemble activity and surface binding of TLL-like variant L4 and WT significantly increased in the application system, while the FAEA-like variants L1 and L2 showed low ensemble activity and surface binding. This can be attributed to the properties of lid mutations as well as the synergistic effect of calcium ions and surfactants. Calcium ions may have lowered the dielectric constant at the water–lipid interface, thus increasing the enzyme’s activity by enhancing the surface binding. L4 and WT had high calcium dependency because of the different lid structures, while L1 and L2 were not sensitive to calcium at all. The presence of surfactant, on the other hand, competed with lipase for binding on the substrate interface, thus decreasing the binding and activity of WT and TLL variants. As the FAEA-like lid conformation is nearly inactive at the water–substrate interface, L1 and L2 displayed relatively lower diffusion coefficients in the application system. It is surprising that although both L4 and WT had higher binding and longer step lengths in the fast state in the application system, their mobilities on the substrate interface were quite different. L4 displayed a larger diffusion coefficient and higher occupancy of the fast state than the other types of TLLs. Meanwhile, a large part of WT stayed in the low diffusional state, resulting in its relatively low average diffusion coefficient. This indicates calcium ions had little impact on how lipase interacts with the substrate after lipase is recruited, and the presence of surfactant in the application system restricted the movement of WT on the substrate surface.

Direct observation of the function, binding, and mobility of lipase via our SPT assay platform has elucidated how functionally designed mutations in the lid directly regulate the behaviors of TLL in an experimental laundry-like application setup. It is instructive on how to design optimal TLL variants for enzyme-based detergents.

## Figures and Tables

**Figure 3 biomolecules-13-00631-f003:**
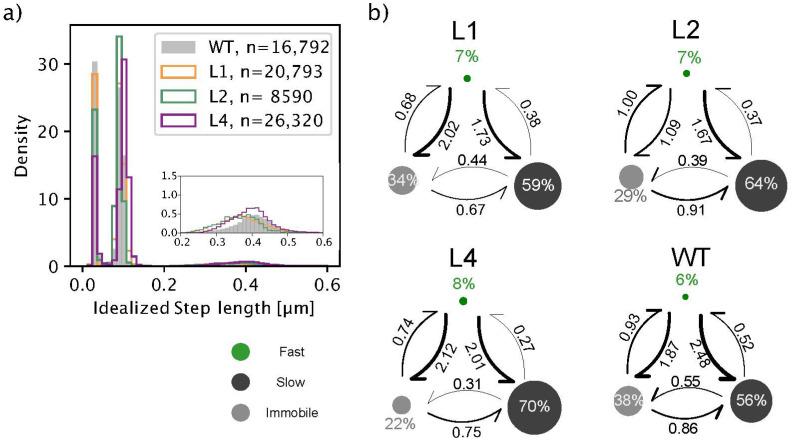
(**a**) Histograms of idealized step length of diffusional states segmented by HMM of step lengths). (**b**) Quantitative model of 3 diffusional states depicting the interconversion rates of TLL and its dependence on mutation in the lid in application conditions. The occupancy percentages are given as the sizes of the circles and the numbers in the circles. The rates between the states are written above the arrows and given in s^−1^.

## Data Availability

All data are available for download upon request.

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
