# Peer review of "Single-Particle Tracking of Thermomyces lanuginosus Lipase Reveals How Mutations in the Lid Region Remodel Its Diffusion"

_biomolecules, 2023, doi:10.3390/biom13040631_

Round 1
Reviewer 1 Report
Comments
MS no.: biomolecules-2244686
The authors utilized single particle tracking method to reveal structure-function relationship in Thermomyces lanuginosus lipase lid and its diffusional behavior. After carefully reviewing, this paper could be accepted after minor revision for publication on Biomolecules.
Major point
1. The WT lipase and its variants (L1, L2, and L4) have different diffusional behavior resulted from lid difference. In the sections of Abstract, Results and Discussion, and Conclusions, the author should point out what concrete structure changes on variants affected their behavioral differences, not only displayed the structures in Fig 1d and e.
2. The writing needs a deep improvement.
Minor points
1. “Thermomyces lanuginosus” and other latin names like “Aspergillus niger” and “Aspergillus terreus” should be italicized throughout the manuscript.
2. Why the same company named by two different names? “Novezymes” on Line 89 Vs. “Novozyme” on Line 95.
3. Page 3, Line 118: What’s the unit for number “1200”?
4. Page 3, Line 141: “CaCl2” should be “CaCl2”.
5. Page 3, Line 143-144: Plz check, here “~ 0.5 nM”, is really using “nM” as unit?
6. Page 4, Line 168: “Ea” should be “Ea” like in the equation on Line 170.
7. Page 6, Line 230: “assays This highlights” should be “assays. This highlights”.
8. Page 9, Line 372: The Doi “https://doi.org/10.1002/(SICI)1521-3773(19980703)37:12<1608::AID-ANIE1608>3.0.CO;2-V” here is missing.
Reviewer 2 Report
In this manuscript, the authors have shown the importance of lid of lipase by designing its different mutants and single particle tracking as a technique. They have unraveled different diffusion states and the transition kinetics of different variants using Hidden Markov Modeling. They find the diffusivity of the enzyme and its variants play an important role for their activity. This study could be helpful in designing the better variants of lipases.
I would recommend accepting the manuscript after minor revision.
Minor comments
1. The correlation between diffusion and enzyme activity should be stated clearly in the abstract.
2. What is the definition of immobile state? The authors should clarify this.
3. What is the localization error? The authors should include localization error distribution for all cases like WT, L1, L2, L3 and L4 in the SI.
4. In Fig. S1, the author should mention the units of X and Y axis parameter. Figure legend is also not clear.
5. In Fig. S4, the authors should represent the X-axis in sec/ms, not in frame.
6. In Fig. S4, for the fast to slow and fast to immobile, the author should try fitting by lowering the bin size. Exponential fitting using only 2 or 3 data points is not convincing.
Reviewer 3 Report
with the article “Single Particle Tracking of Thermomyces lanuginosus lipase reveals how mutations in the lid region remodels its diffusion”, Iversen et al want to understand the function of TLL lid mutations in application conditions. I appreciate the applicative aim of this study.
Here are some comments.
in the title, being a microorganism, it’s important to write the name of T. lanuginosus in italics (and even always in the manuscript, and also per Aspergillus sp).
line 39, add references about the needs of activation of TLL
line 97, the only citation of previous work is insufficient. Please, provide also a short method.
line 104, TIRS buffer instad of TRIS
in the methods section, I’d like to have more information about the rational design of the TLL variants (even if made by Novozymes). Why did you decide on these specific changes in the lid?
Lines 197-199, the diffusion coefficient allowed the evaluation of catalytic activity. This part needs a further explanation about how this has been done.
The variants are actually three. WT is not a variant.
Where is L3?
Paragraph 3.2. WT is called "mutant" but it is a wild type. This is confused
Reviewer 4 Report
The manuscript is well written and the presentation of the results demonstrate a thorough investigation of the lid in Thermomyces lanuginosus, (which must be written in italics). The manuscript may be published after this correction both in the title and in the text.
